# Are Ethiopian diabetic patients protected from financial hardship?

**Gebeyehu Tsega** *, **Gebremariam Getaneh, Getasew Taddesse**

Department of Health Systems Management and Health Economics, School of Public Health, Bahir Dar University, Bahir Dar, Ethiopia

* gebishts@gmail.com

**Data Availability Statement:** All relevant data are within the paper and its Supporting information files.

**Funding:** The author(s) received no specific funding for this work.

## Abstract

### Background

Globally, diabetes mellitus exerts an economic burden on patients and their families. However, the economic burden of diabetes mellitus and its associated factors were not well studied in Ethiopia. Therefore, the aim of this study is to assess the economic burden of diabetes mellitus and its associated factors among diabetic patients in public hospitals of Bahir Dar city administration, Ethiopia.

### Methods

Across sectional study was conducted on 422 diabetic patients. The patients were selected by simple random sampling method. The prevalence-based model was used to estimate the costs on patients' perspective. Bottom up and human capital approaches were used to estimate the direct and indirect costs of the patients respectively. Wealth index was constructed using principal component analysis by SPSS. Forty percent of nonfood threshold level was used to measure catastrophic diabetic care expenditure of diabetic patients. Whereas, the World Bank poverty line (the $1.90-a-day poverty line) was used to measure impoverishment of patients due to expenses of diabetes mellitus care. Data were entered by Epi data version 3.1and exported to SPSS version 23 for analysis. Simple and multiple logistic regressions were used.

### Results

Four hundred one respondents were interviewed with response rate of 95%. We found that 239 (59.6%) diabetic patients incurred catastrophic diabetic care expenditure at 40% nonfood threshold level. Whereas, 20 (5%) diabetic patients were impoverished by diabetic care spending at the $1.90-a-day poverty line. Educational status of respondent, educational status of the head of household, occupation and wealth status were statistically associated with the catastrophic diabetic care expenditure.

### Conclusions

The study revealed that the economic burden of diabetic care is very disastrous among the less privileged populations: the less educated, the poorest and unemployed. Therefore, all

**Competing interests:** The authors have declared
that no competing interests exist.

concerned stakeholders should design ways that can reduce the financial hardship of dia-
betic care among diabetic patients.

## Introduction

Diabetes Mellitus (DM) is a chronic disease that occurs when there is a raised level of glucose
in the blood [1–3]. DM is a major global health threat [4–6]. It exerts a heavy economic burden
on society. The burden is related to direct costs, indirect costs, and intangible costs of diabetes
mellitus that incurred by patients [7,8].

Nowadays, high attention is given for protecting people from financial hardship to access
health care in the international and national strategic plans. Financial risk protection means
that everyone can obtain the health care services they need without experiencing financial
hardship. It is a key health system objective and one of the targets in the Sustainable Develop-
ment Goals (SDGs), specifically SDG3. Financial risk protection is also the target of Health
Sector Transformational Plan (HSTP) of Ethiopia [9]. However, literature stated that150 mil-
lion people faced catastrophic health expenditure (at 40% threshold level) and one hundred
million people are pushed in to poverty annually due to their out-of-pocket payments (OOPs)
for health care at the $1.90-a-day poverty line threshold. Ninety seven percent (97%) of the
impoverished population are found in Asia and Africa. Only in Africa region, 11 million peo-
ple per year become impoverished as result of high out-of-pocket payments. According to the
latest global monitoring report, catastrophic health spending as measured by SDG indicator
3.8.2 will continue to increase until 2030 [10–12].

In Ethiopia, out of pocket health spending amounted to 31% of the total health expenditure,
which is higher than that of the global recommended target, 20%, [13]. People with chronic
diseases like diabetes disproportionally suffered from financial hardship, catastrophic health-
care expenditure and impoverishment, especially in the low-income countries. For diabetic
patients in low-income countries like Ethiopia, financial hardship is very disastrous as the
poor or low socioeconomic groups may be forced to forego other vital needs such as dietary
diversity in order to attain health services. Besides socioeconomic status, sex, educational sta-
tus, household size, gender of household size, place of residence and the presence of vulnerable
groups were determinant factors of catastrophic health expenditure (CHE) and impoverish-
ment as the result of health care spending [14–16].

Evidence on financial hardship of diabetic care is essential to improve equity of diabetic
health care among diabetic patients. It is also essential to remove financial barriers that are
slowing down access to diabetic health care among diabetic patients in Ethiopia [17]. However,
the economic burden and its associated factors of diabetes mellitus on diabetic patients were
not well studied in Ethiopia. Therefore, the aim of this study is to assess the economic burden
and its associated factors of diabetes mellitus on patients in public hospitals found in Bahir
Dar city administration, Ethiopia.

## Methods and materials

### Study design, setting and participants

Across sectional study was conducted from September 15/2019 to December 30/2019 in public
hospitals found in Bahir Dar city administration, Ethiopia. Bahir Dar city has three public hos-
pitals (FelgeHiwot comprehensive specialized hospital, AdisAlem primary hospital and Tibebe
Gihon hospital). These three hospitals are serving about 5.5 million people. Since Tibebe

Gihon hospital has given services less than one year at the time of data collection, it was excluded from the study. About 1300 diabetic patients have follow up in FelgeHiwot comprehensive specialized hospital and AdisAlem primary hospital.

Diabetic patients with 14 or more years old, who are receiving services in the selected hospitals, were candidates for the study. However, community based health insurance enrolled diabetic patients were excluded from the study.

## Sample size and sampling method

The number of study participants included in this study was determined by using the single population proportion formula. We used expected proportion (p) of the study participants who have catastrophic diabetic care expenditure of 50%, marginal error (d) 5% and confidence interval of 95%. Therefore, by considering 10% non-response rate, the final sample size was 422. Moreover, the 422study participants were randomly and proportionally recruited from the two public hospitals. Hence, we recruited 325 diabetic patients from FelegeHiwot comprehensive specialized hospital and97 diabetic patients from AddisAlem primary hospital by using simple random sampling method with computer generated random number.

**Measurement and data collection.** The study used a structured questionnaire, which consisted of questions on sociodemographic and socioeconomic characteristics, household expenditure as well as health care payments of DM (Additional file 1). The questionnaire was translated to local, Amharic, language (Additional file 2). Pretest was conducted and modifications were made accordingly. Trained data collectors, three BSC Nurses, have collected the data from study participants through face-to-face exit interview. The supervisor and principal investigator closely monitored the data collection process on daily base.

Participants were asked about their direct medical costs (medical card, laboratory test, insulin syringe and medications), direct non-medical costs (transportation, cafeteria, lodging) and indirect costs (patients and caregivers' time loss) during exit interview.

Household consumption expenditure related data were collected by asking respondents/ care givers for monthly estimates of amounts spent on food, housing, electricity, water, diabetic healthcare and others for the 12 months preceding the survey. Respondents were also asked about the availability of durable consumer goods such as radio, television, mobile phone, bicycle, farm equipment and agricultural land, livestock and also the amount of cereals and grains they collected over the preceding year.

**Method of cost estimation.** The costing method of the study was based on the patients' perspective. Micro-costing(bottom-up) approach was used to estimate the direct costs of diabetes mellitus. Whereas, human capital approaches, in terms of productivity time losses, was employed to estimate the indirect costs of DM. Regarding to the time frame, prevalence based model was used. Costs were estimated for each patient for the 12 months preceding the survey [18,19].

We included costs for outpatient visits, drugs and laboratory tests to estimate the direct health costs of DM. Whereas, costs for transportation, meals and accommodation/lodging during outpatient visits, for both the patient and the accompanied person, were included to estimate the direct none medical costs of DM.

We calculated the indirect cost in terms of lost days of productivity for the patients and/or caregivers. In this category of cost, we included earnings lost because of travel to outpatient visits and those due to absences from work because of illness related to DM [20–22].

For formal employed workers (payroll paid) monetary value of lost days has been calculated by multiplying number of lost days with reported personal daily income (monthly income divided by 30). For farmers their reported annual income was determined by converting the

cereals and grains they owned per year into monetary value using approximate market value of the year and then divided by 12 to get monthly income. Then, the monthly income of the farmer was divided by 30 days to get individual daily income and this was multiplied by lost days.

**Measuring catastrophic health expenditure of diabetic care.** The Wag staff and van Doorslaer method was used to measure the CHE and impoverishment. To calculate the catastrophic head count of diabetic care which is the percentage of patients/clients incurring catastrophic expenditures, we defined DMxi to be expenditure of the i th patient to receive care for DM for patient i, $x_i$ total expenditure for patient i, and f(x) food expenditures. The diabetic patient is said to have incurred catastrophic diabetic care payments, if [DMx$_i$/(x$_i$-f(x))] $^*$100 exceeds a specified threshold, z (in our case 40% of nonfood threshold level was used for analytical statistics). But the choice regarding the threshold to use in determining catastrophic health expenditure is arbitrary and has typically varied between 10 and 40% of threshold level [23].

**Measuring incidence and intensity of catastrophic diabetic care payments.** The headcount (H) is the given by:-

$$H = \frac{1}{N}\sum_{i=1}^{N} E_i \tag{1}$$

Where N is the sample size and $E_i$ equals 1 if DMx$_i$/[x$_i$-f(x)] $>$ z and zero otherwise.

The headcount does not reflect the amount by which households exceed the threshold. We therefore used the catastrophic expenditure overshoot, which captures the average degree by which health expenditures (as a proportion of total expenditure or non-food expenditure) exceed the threshold z. The overall overshoot (O) is given by:-

$$O = \frac{1}{N}\sum_{i=1}^{N} O_i \tag{2}$$

Where $O_i = E_i ((DMx_i/x_i\text{-}f(x)) - z)$.

The incidence (headcount) and the intensity (overshoot) of catastrophic expenditures are related through the mean positive overshoot (MPO) which captures the intensity of occurrence of catastrophic expenditures defined as overshoot divided by headcount:

$$MPO = \frac{O}{H}; \; O = H * MPO \tag{3}$$

**Measuring impoverishment.** Wag staff and van Doorslaer also describe methods to adjust poverty measures on the basis of household expenditure net of OOP spending on health care [23]. The three measures of poverty include;

1. Poverty head count, which is the proportion of households living below the poverty line (PL);

$$H_{pov}^{pre} = {}^1\!/_N\sum_{i=1}^{N} P_i^{pre} = \mu P pre \tag{4}$$

Where $H_{pov}^{pre}$ is poverty headcount before health payment and P$_i^{pre}$ = 1 if X$_i$> PL and zero otherwise.

2. Poverty gap, referring to the aggregate of all short falls from the poverty line;

$$G_{pov}^{pre} = {}^1/_N \sum_{i=1}^{N} g_i^{pre} = gpre \qquad (5)$$

Where $G_{pov}^{pre}$ is prepayment poverty gap, $g_i^{pre}$ = PL-$X_i$ if PL>$X_i$ and zero otherwise.

3. Normalized poverty gap ($NG_{pov}^{pre}$) or poverty gap index is obtained by dividing the poverty gap by the poverty line.

$$NG_{pov}^{pre} = \frac{G_{pov}^{pre}}{PL} \qquad (6)$$

Calculating the three poverty measures requires setting a poverty line and assessing the extent to which health care payments push households below the poverty line. The World Bank poverty line1.9 USD dollar per person per day converted to Ethiopian birr based on average exchange rate (1USDdollar = 28.18 ETB) of September 2018-August 2019 was used to estimate poverty levels before and after healthcare payments. Replacing all the pre-payment superscripts, 'pre' by the superscript 'post' gives the analogous post-payment measurement.

The measures of poverty impact (PI$^H$) of health payments are then simply defined as the difference between the pre-payment and post-payment measures, i.e.

$$PI^H = H_{pov}^{post} - H_{pov}^{pre} \qquad (7)$$

Where $H_{pov}^{post}$ and $H_{pov}^{pre}$ are post and pre health payments poverty incidence respectively.

**Data processing and analysis.** The data were checked for completeness. Then, data were coded, organized and entered into Epi-data Version 3.1 and exported to SPSS version 23 software for cleaning and analysis. Descriptive statistical analysis, simple and multiple logistic regressions were conducted. All variables with p-value less than 0.25 in bi-variable analysis were considered as candidates for multivariable logistic regressions analysis. Adjusted odds ratio (AOR) with 95% CI was used to identify significantly associated variables. Wealth index was constructed using principal component analysis based on housing condition, water source and household durable assets.

**Ethical clearance.** Ethical clearance was obtained from Institutional Review Board (IRB) of Bihar Dar University, School of Public Health on January 24/2019. The ethics approval reference number is RCS/010/2019. A formal letter, from the school was submitted to each concerned bodies to obtain their co-operation. Explanatory letter was added to each questionnaire to maintain participants rights, also, all patients asked to participate in the study and received full explanations about the research purposes. Respect, anonymity and confidentiality were given and maintained by consent form for each participants and the liberty to withdraw at any stage of the interview and their participation was undergo to any pressure. Then, written informed consent was obtained from the participant as per the Institutional review board (IRB) approval.

## Results

### Sociodemographic and socio economic characteristics of respondents

Four hundred one respondents were interviewed with response rate of 95%. The age of respondents ranges from 15–80 years with mean of 43.27 (SD ±14.5). Out of the total 401 respondents, 235(58.6%) were males whereas most of the respondents (74.6%) were orthodox Christians. The majority of the respondents were ethnically Amhara (83.8%). More than half

**Table 1. Socio-demographic and socio-economic characteristics of diabetic patients in Bahir Dar city administration public hospitals, Ethiopia, 2019.**

| Variables | | Frequency | % |
|---|---|---|---|
| Sex | Male | 235 | 58.6 |
| | Female | 166 | 41.4 |
| Age in years | 15–30 | 99 | 24.7 |
| | 31–45 | 113 | 28.2 |
| | 46–60 | 115 | 38.7 |
| | >60 | 34 | 8.5 |
| Ethnicity | Amhara | 336 | 83.8 |
| | Others[1] | 65 | 16.2 |
| Religion | Orthodox | 299 | 74.6 |
| | Others[2] | 102 | 25.4 |
| Marital status | Single | 90 | 22.4 |
| | Married | 244 | 60.9 |
| | Widowed/divorced | 67 | 16.7 |
| Educational status | No formal education | 183 | 45.6 |
| | Primary education | 51 | 12.7 |
| | Above primary | 167 | 41.7 |
| Occupational status | Unemployed | 125 | 31.1 |
| | Employed (Payroll paid) | 111 | 27.7 |
| | Farmer | 84 | 21 |
| | Merchant | 81 | 20.2 |
| Place of residence | Urban | 289 | 72.1 |
| | Rural | 112 | 27.9 |
| Households monthly income | < = 2,500 ETB | 98 | 24.4 |
| | 2,501–5,000 | 196 | 49.0 |
| | 5,001–1,000 | 84 | 20.9 |
| | >10,000 | 23 | 5.7 |
| Households socioeconomic status | 1st quintile | 81 | 20.2 |
| | 2nd quintile | 84 | 20.9 |
| | 3rd quintile | 79 | 19.7 |
| | 4th quintile | 68 | 17 |
| | 5th quintile | 89 | 22.2 |

[1] Oromo, Tigray, Gumuz;

[2] Muslim, Protestant, Catholic.

of the respondents were married-244(60.9%). Nearly half of the respondents-183(45.6%) had no formal education. One hundred and eleven (27.7%) of the respondents were payroll paid and 84(21%) were farmers. Household's family size ranges from one up to ten with the mean size of 4.24 (SD±1.82). The percentages of economically dependent members accounted for 11.81% of total households' size. Majority of the respondents 289(72.1%) were urban dweller. The wealth status of the households was classified in to five categories from first quintile to fifth quintile, and 22.2% of households were on fifth quintile. The mean monthly income of the respondents was 4710 (Table 1).

## Clinical characteristics of diabetic mellitus and related issues

About 243(60.6%) respondents were type II Diabetic patients. The mean duration of illness of respondents living with diabetes mellitus was 7.72(SD±5.43) years. One hundred fifty eight

(39.4%) of respondents had monthly follow upwhereas112 (27.9%) had follow up every three months. Majority of respondents,287(71.6%) were stressed because of being diabetic patient. Among the respondents, 170(42.4%) were doing preventive measures to control blood sugar level.

## Cost of diabetes mellitus treatment, household expenditure

The mean total monthly household expenditure was 3568.40ETB with (SD ±2077.4). The average monthly household's food, nonfood and health expenditures were 2285.49ETB with (SD ±1446.82), 1282.92ETB (SD ±1173.27) and 505 ETB (SD± 400.94), respectively.

The mean monthly direct medical cost was 382.48 ETB (SD± 324.46). As reported by 193 (48.13%) respondents, the mean direct cost of insulin and insulin syringes were 190.98ETB (SD± 110.64) and 48.4ETB (SD ±51.63) respectively. The mean monthly direct medical cost for oral anti diabetic medication users, 223(55.61%) was 327.94ETB (SD ±124.01); and for laboratory service the average cost was16.56ETB (SD ±24.97) as reported by all respondents (Table 2).

Direct medical cost and direct non-medical costs accounted 75.74% and 9.65% of health expenditures, respectively. The mean monthly indirect cost calculated by taking lost days for both the patient and care givers and their estimated daily income into consideration was 73.77ETB(SD ±112.57); which accounts 14.61% of monthly health expenditure (Table 2).

## Coping strategies

Nearly two-thirds (60.6%) of respondents used their own money (savings and salary), while 21.2% from family/relative support, 13.5% by selling assets and 3% by borrowing from someone to cope the diabetes care payment. About 50.63% and 24.67% households faced CHE cope cost of treatment for DM by drawing savings and relative/family support respectively. Moreover, 85% and 15% of impoverished households were tried to cope diabetes treatment care by drawing saving and by selling household assets, respectively.

**Table 2. Expenditures of diabetic patients in Bahir Dar city administration public hospitals, Ethiopia, 2019.**

| Variable | N | Mean(ETB) | Std. Dev. | median |
|---|---|---|---|---|
| Household costs per month | | | | |
| Total household expenditure | 401 | 3568.4 | 2077.4 | 3095.5 |
| Household food expenditure | 401 | 2285.49 | 1446.822 | 2000 |
| Nonfood household expenditure | 401 | 1282.92 | 1173.27 | 925 |
| Direct medical cost per patient per month | 401 | 382.48 | 324.46 | 346.66 |
| Insulin | 193 | 190.98 | 110.64 | 166.67 |
| Insulin syringe | 193 | 48.4 | 51.63 | 40 |
| Laboratory test | 401 | 16.56 | 24.97 | 10 |
| Oral anti diabetics | 223 | 327.94 | 124.01 | 400 |
| Medical card | 401 | 6.2 | 4.05 | 5 |
| Direct non-medical cost (monthly) | 401 | 48.75 | 91.19 | 22.5 |
| Transport cost | 354 | 28.16 | 51.94 | 13.33 |
| Food cost during hospital visit | 193 | 38.17 | 49.36 | 25 |
| Lodging cost during hospital visit | 32 | 69.66 | 76.45 | 50 |
| Indirect monthly cost(due to lost days) | 401 | 73.77 | 112.57 | 44.44 |
| Total monthly health payment of DM | 401 | 505 | 400.94 | 444.44 |

Note: All monetary values are presented in Ethiopian birr, N—number of observations. Std. Dev.–Standard deviation.

## Catastrophic health expenditure and impoverishment

The proportion of catastrophic health expenditure among diabetic patients using 40% (non-food) threshold was 59.6%. Among respondents who faced catastrophic health expenditure, 145 (60.67%) were urban dwellers, 139(58.16%) were married, 94(39.33%) were in the age range of 46–60 years, 136 (57%) had no formal education, 63(26.36%) were in the third quintile wealth status and 30(12.55%) were in fifth quintile wealth status (Table 3).

Three hundred and four (75.8%) of the respondents were poor before paying for diabetes care whereas 20(5%) of them were impoverished after paying for diabetes care. Among impoverished respondents, 14(70%) were male participants, 15(75%) were educated above primary level. Regarding to wealth status, 9.1%, 5.3%, 4.3%, 3.8% and 3% of impoverished respondents were in the 2nd, 3rd, 5th, 1st and 4th quintiles respectively (Table 3).

The incidence (headcount) and intensity (overshoot) of catastrophic diabetic expenditures were59.6 and 23.46% respectively. On the other hand, the proportion of mean positive over shoot (MPO) was39.36% (Table 4).

Impoverishment was estimated by calculating poverty levels using consumption expenditures before and after paying for diabetic care. Both the headcount and the poverty gap were calculated based on the World Bank poverty line1.9 USD which is equivalent to ETB 1606.26 per person per month. About 75.8% of respondents were living below poverty line before paying for diabetic care. After paying for diabetic care, the headcount increased by 5%. The average shortfall from the poverty line (the poverty gap) were ETB 1960.8(69.58 USD) and ETB 2336.44(82.91USD) before and after accounting for diabetic care payments respectively. There was an increase in poverty gap of ETB 375.64(13.33 USD) after diabetic care payment. Whereas, the mean positive poverty gap before and after diabetic care payments were 45.66% and 52.5% respectively (Table 5).

## Factors associated with catastrophic health expenditure

Occupation, educational status of respondents, educational status of household head and wealth status were independent predictors of catastrophic expenditure for diabetic care. However, sex of the respondents, religion of respondents, place of residence, marital status, sex of household head, presence of under five children, frequency of follow up and source of medication were not independent predictors of catastrophic expenditure for diabetic care (Table 6).

Diabetic patients who attended above primary school were 68.8% (AOR = 0.312; 95%CI: 0.125, 0.776) less likely to have catastrophic expenditure for diabetic care as compared to those with no formal education. Diabetic patients whose households led by heads with primary education were 65.7% (AOR = 0.343; 95%CI: 0.134, 0.875) less likely to have catastrophic expenditure for diabetic care as compared to those with households led by a head with no formal education (Table 6).

Diabetic patients who were in 2nd and 3rd wealth quintiles were 2.4 times (AOR = 2.417; CI: 1.079, 5.413) and 2.7 times (AOR = 2.744; CI: 1.161, 6.187) more likely to encounter catastrophic expenditure for diabetic care respectively as compared with that of diabetic patients in the 5th wealth quintile (Table 6).

Diabetic patients who were formal employees and merchants were 54.8% (AOR = 0.452; CI: 0.225, 0.906) and 58.4% (AOR = 0.416; CI: 0.206, 0.827) less likely to catastrophic expenditure for diabetic care respectively as compared to those with unemployed diabetic patients (Table 6).

## Discussion

This study aimed to assess the economic burden of health expenditure in diabetic patients in public hospitals of Bahir Dar city, North West Ethiopia. The study showed that the average

**Table 3. Catastrophic health expenditure and impoverishment among diabetic patients in public hospitals of Bahir Dar city administration, Ethiopia, 2019.**

| Variables | Categories | Catastrophic health care expenditure of DM | | Impoverishment | |
|---|---|---|---|---|---|
| | | No | Yes | No | Yes |
| sex | Male | 108 | 127 | 221 | 14 |
| | Female | 54 | 112 | 160 | 6 |
| age | 15–30 | 37 | 62 | 91 | 8 |
| | 31–45 | 53 | 60 | 108 | 5 |
| | 46–60 | 61 | 94 | 148 | 7 |
| | >60 | 11 | 23 | 34 | 0 |
| Marital status | Single | 37 | 53 | 82 | 8 |
| | Married | 105 | 139 | 235 | 9 |
| | widowed/Divorced | 20 | 47 | 64 | 3 |
| Religion | Orthodox | 113 | 186 | 287 | 12 |
| | [1]Others | 49 | 53 | 94 | 8 |
| Ethnicity | Amhara | 132 | 204 | 320 | 16 |
| | [2]Others | 30 | 35 | 61 | 4 |
| Educational status | No formal education | 34 | 136 | 166 | 4 |
| | Primary | 19 | 33 | 51 | 1 |
| | Above primary | 109 | 70 | 164 | 15 |
| Occupational status | Unemployed | 38 | 87 | 123 | 2 |
| | Payroll paid | 70 | 41 | 99 | 12 |
| | Farmer | 42 | 39 | 78 | 3 |
| | Merchant | 12 | 72 | 81 | 3 |
| Place of residence | Urban | 144 | 145 | 275 | 14 |
| | Rural | 18 | 94 | 106 | 6 |
| Sex of household head | Male | 143 | 189 | 317 | 15 |
| | Female | 19 | 50 | 64 | 5 |
| Educational status of household head | No formal education | 39 | 144 | 178 | 5 |
| | Primary | 23 | 28 | 50 | 1 |
| | Above primary | 100 | 67 | 153 | 14 |
| Frequency of follow up | Monthly | 56 | 102 | 150 | 8 |
| | Every two months | 34 | 35 | 64 | 5 |
| | Every three months | 43 | 69 | 106 | 6 |
| | Every four months | 29 | 33 | 61 | 1 |
| Socioeconomic status based on wealth index | 1st quintile | 34 | 47 | 78 | 3 |
| | 2nd quintile | 24 | 60 | 77 | 7 |
| | 3rd quintile | 16 | 63 | 75 | 4 |
| | 4th quintile | 29 | 39 | 66 | 2 |
| | 5th quintile | 59 | 30 | 85 | 4 |

[1]Muslim, Catholic and Protestant;

[2] Tigray, Gumuz, and Oromo.

**Table 4. Extent and intensity of catastrophic health expenditure at variable threshold levels of diabetic patients in public hospitals of Bahir Dar city administration, Ethiopia, 2019.**

| | Catastrophic health expenditure | | | | | | | |
|---|---|---|---|---|---|---|---|---|
| | As a share of total monthly expenditure | | | | As a share nonfood monthly expenditure | | | |
| | 10(%) | 20(%) | 30(%) | 40(%) | 10(%) | 20(%) | 30(%) | 40(%) |
| Catastrophic headcount (%) | 74.3 | 28.7 | 9.7 | 4.2 | 98 | 85.8 | 74.8 | 59.6 |
| Catastrophic overshoot (%) | 8.05 | 3.16 | 1.4 | 1.19 | 47.37 | 38.19 | 30.16 | 23.46 |
| Mean positive gap (%) | 10.83 | 11.01 | 14.43 | 28.33 | 48.33 | 44.51 | 40.32 | 39.36 |

monthly diabetic care expenditure is 505 ETB (17.92 USD). It is lower than that of a study done on catastrophic health care expenditure in Ethiopia (610ETB) [24]. The possible explanation for this difference might be due to the fact that the previous study incorporated catastrophic health care expenditure as results of all diseases while the current study was conducted on catastrophic expenditure for diabetic care.

In the current study, the direct medical cost of diabetic care accounted 75.74% of the total cost of diabetic care. This finding is in line with that of studies done in Ghana (78%) and Nepal (75.8%) [25,26]. But it is lower than that of a study done in china (90.9%) [27]. This difference might be due to the fact that the contexts of the studies are different in terms of socio economic status.

In this study at 10% threshold of total household expenditure, the incidence (headcount) of catastrophic health expenditure was 74.3%. This result is higher than that of a previous study done in Ethiopia in which the incidence was 24% [28]. The difference might be due to the fact that in the current study the study participants were diabetic patients which are prone to catastrophic health expenditure.

In the present study, catastrophic expenditure of diabetic care at 40% threshold was 59.6%. This is higher than that of previous studies done in South Africa (6%) and China (13.8%) [15,29]. The difference might be due to the fact that all study participants of the previous studies were urban dwellers and insured participants were included in the studies. Moreover, South Africa and China have better socioeconomic status than that of Ethiopia. The other difference might be due to the fact that in current study, cafeteria costs and care givers costs were included as opposed to that of South Africa and China. The incidence of catastrophic expenditure of diabetic care is also higher than that of a previous study done in 35 developing countries (17.8%) [30]. The difference might be due to different study contexts.

The catastrophic overshoot and MPO at 40% non-food threshold were 23.46% and 39.36%, respectively. The overshoot implied that on average all diabetic patients included in the study had invested 63.46% (23.46%+40%) of their monthly expenditure on diabetic care. Whereas,

**Table 5. Average monthly poverty headcount and gap before and after paying for diabetic among diabetic patients in public hospitals of Bahir Dar City administration, Ethiopia, 2019.**

| Impoverishment status(monthly) | | | |
|---|---|---|---|
| Poverty headcount | | Poverty gap (ETB (%)) | |
| Prepayment headcount | 75.8% | Prepayment poverty gap | 1960.8(45.66%) |
| Post payment headcount | 80.8% | Post payment poverty gap | 2336.44(52.5% |
| Percentage change | 5% | Point change | 375.64 (19.16%) |
| | | Prepayment poverty gap index | 34.61% |
| | | Post payment poverty gap index | 42.42% |

**Table 6. Logistic regression results on predictors of catastrophic expenditure of diabetic care among diabetes mellitus patients in public hospitals of Bahir Dar city administration, Ethiopia, 2019.**

| | | CHE | | | |
|---|---|---|---|---|---|
| **variables** | | **No** | **Yes** | **COR(95% CI)** | **AOR(95% CI)** |
| Sex | Male | 108 | 127 | 1 | 1 |
| | female | 54 | 112 | 1.764(1.166, 2.668) | 0.902(0.510,1.596) |
| Religion | Orthodox | 113 | 186 | 1 | 1 |
| | [1]Others | 49 | 53 | 0.657(0.418, 1.034) | 0.912(0.519,1.603) |
| occupation | Unemployed | 38 | 87 | 1 | 1 |
| | Payroll paid | 70 | 41 | 0.256(0.149, 0.440) | 0.453(0.226,0.907)* |
| | Merchant | 42 | 39 | 0.406(0.227, 0.724 | 0.416(0.208,0.833)* |
| | Farmer | 12 | 72 | 2.621(1.275, 5.385) | 0.651(0.206,2.064) |
| Educational status | No formal education | 34 | 136 | 1 | 1 |
| | Primary education | 19 | 33 | 0.434(0.220, 0.855) | 1.104(0.411,2.966) |
| | Above primary education | 109 | 70 | 0.161(0.099, 0.260 | 0.310(0.125,0.771)* |
| Place of residence | Urban | 144 | 145 | 1 | 1 |
| | Rural | 18 | 94 | 5.186(2.979, 9.029) | 2.138(0.875,5.224) |
| Marital status | Single | 37 | 53 | 1 | 1 |
| | Married | 105 | 139 | 0.924(0.566, 1.509) | 1.003(0.522,1.926) |
| | Divorced/widowed | 20 | 47 | 1.641(0.839, 3.209) | 0.754(0.303,1.877) |
| Sex of household head | Male | 143 | 189 | 1 | 1 |
| | female | 19 | 50 | 1.991(1.125, 3.525) | 1.879(0.788,4.481) |
| Education of household head | No formal education | 39 | 144 | 1 | 1 |
| | Primary education | 23 | 28 | 0.330(0.171, 0.635) | 0.347(0.136,0.884)* |
| | Above primary education | 100 | 67 | 0.181(0.113, 0.290) | 0.959(0.402,2.286) |
| U-5 children | No | 92 | 157 | 1 | 1 |
| | Yes | 70 | 82 | 0.686(0.456, 1.034) | 0.641(0.385,1.078) |
| | Monthly | 56 | 102 | 1 | 1 |
| Frequency of follow up visit | Two monthly | 34 | 35 | 0.565(0.318, 1.003) | 0.604(0.299,1.220) |
| | Three monthly | 43 | 69 | 0.881(0.534,1.454) | 0.773(0.419,1.425) |
| | Four monthly | 29 | 33 | 0.625(0.344, 1.134) | 0.672(0.325,1.388) |
| Source of medication | Government | 83 | 173 | 1 | 1 |
| | Non-government | 79 | 66 | 0.401(0.264, 0.609) | 0.664(0.393,1.123) |
| Socioeconomic status | 1st quintile | 34 | 47 | 2.719(1.458, 5.068) | 1.482(0.683,3.217) |
| | 2nd quintile | 24 | 60 | 4.917(2.577, 9.380) | 2.448(1.094,5.475)* |
| | 3rd quintile | 16 | 63 | 7.744(3.834, 15.641) | 2.715(1.151,6.409)* |
| | 4th quintile | 29 | 39 | 2.645(1.379, 5.073) | 1.868(0.903,3.863) |
| | 5th quintile | 59 | 30 | 1 | 1 |

* = Significant at p<0.05.

the MPO indicated that only those diabetic patients with catastrophic expenditure of diabetic care had invested 79.36% (39.36%+40%) of their monthly expenditure on diabetic care.

Regarding to impoverishment, both poverty headcount and poverty gap became higher after payment for the diabetic care. In this study, we found that 5% of diabetic patients fell into poverty after the diabetic care payment. This finding is in line with that of previous studies done in South Africa (4%), Ethiopia (5.8%) and Kenya (4% and 5.4%) [15,28,31,32].

The average shortfall from poverty line, poverty gap, following diabetic care payment was substantial. On average, the diabetic care expenditure increased the poverty gap of diabetic

patients by 19.16% as compared to before diabetic care payment. This finding highlights that the diabetic care expenditure severely affected the pre-payment poor diabetic patients. The proportion of prepayment poverty gap index and post payment poverty gap index were 34.61% and 42.42%, respectively. This means, on average the diabetic patients werefar below the poverty line by 34.61% and 42.42% before and after diabetic care payment, respectively.

Diabetic patients which are in 1st wealth quintile had lowest incidence of impoverishment (3.8%) and those in the 2nd and 3rd quintiles had 9.1% and 5.3%incidence respectively. This finding is in line with that of previous study done in Kenya [32]. The reason for the lowest incidence for the 1st quintile can be explained by the fact that households in this quintile are already poor i.e. 90.1% of households are below the poverty line, even before diabetic care payment.

The current study revealed that catastrophic diabetic care expenditure, which was measured at 40% threshold (nonfood share), was affected by occupation, educational status of respondents, educational status of household heads and wealth status.

The result of this study revealed that diabetic patients which were in the higher wealth quintile have low probability of incurring catastrophic diabetic care expenditure or vice versa. This finding is consistent with that of previous studies done in South Africa [15,33]. Diabetic patients whose household headed by a person with a lower level of education were far more likely to encounter catastrophic diabetic care expenditure. This finding is consistent with that of a study done in Latvia [34].

The current study has its own limitation. The finding of the study may be affected by recall bias due to the fact that the respondents may not remember the information related to their past diabetic care payment during interviewing through retrospective questions.

## Conclusions

The study showed that diabetes mellitus is imposing a significant economic burden to the patients. The current study also revealed that catastrophic diabetic care expenditure was affected by occupation, educational status of respondents, educational status of household heads and wealth status. Therefore, all responsible stakeholders should design ways that can reduce the financial hardship of diabetic care among diabetic patients, specifically the poor, unemployed and uneducated diabetic patients.

## Supporting information

**S1 Table. Clinical characteristics of diabetes mellitus and related issues among diabetic patients having regular follow up at public hospitals of Bahir Dar city administration, North West Ethiopia, 2019.**
(DOCX)

**S2 Table. Coping strategies for diabetes mellitus health care costs among diabetic patients having regular follow up at public hospitals of Bahir Dar city administration, North West Ethiopia, 2019.**
(DOC)

**S1 Text. Survey questionnaire in English.**
(DOCX)

**S2 Text. Local language version questionnaire (Amharic language).**
(DOCX)

## Acknowledgments

We would like to thank Bahir Dar University, study participants, data collectors and supervisors for their contributions for the study. We also thank Felegehiwot and AdisAlem Hospitals for their support during the process of the study.

## Author Contributions

**Conceptualization:** Gebeyehu Tsega, Gebremariam Getaneh, Getasew Taddesse.

**Data curation:** Gebeyehu Tsega, Gebremariam Getaneh, Getasew Taddesse.

**Formal analysis:** Gebeyehu Tsega, Gebremariam Getaneh.

**Funding acquisition:** Gebeyehu Tsega, Gebremariam Getaneh, Getasew Taddesse.

**Investigation:** Gebeyehu Tsega, Gebremariam Getaneh.

**Methodology:** Gebeyehu Tsega, Gebremariam Getaneh, Getasew Taddesse.

**Project administration:** Gebeyehu Tsega, Gebremariam Getaneh, Getasew Taddesse.

**Resources:** Gebeyehu Tsega, Gebremariam Getaneh.

**Software:** Gebeyehu Tsega, Gebremariam Getaneh.

**Supervision:** Gebeyehu Tsega, Getasew Taddesse.

**Validation:** Gebeyehu Tsega.

**Writing – original draft:** Gebeyehu Tsega, Gebremariam Getaneh, Getasew Taddesse.

**Writing – review & editing:** Gebeyehu Tsega, Gebremariam Getaneh, Getasew Taddesse.

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
