## [Decision Letter · Decision Letter 0]

12 Oct 2020

PONE-D-20-25611

Are Ethiopian diabetic patients protected from financial hardship?

PLOS ONE

Dear Dr. Tsega,

Thank you for submitting your manuscript to PLOS ONE. After careful consideration, we feel that it has merit but does not fully meet PLOS ONE’s publication criteria as it currently stands. Therefore, we invite you to submit a revised version of the manuscript that addresses the points raised during the review process.

We look forward to receiving your revised manuscript.

Kind regards,

Khurshid Alam, Ph. D.

Academic Editor

PLOS ONE

Journal Requirements:

3. Please include additional information regarding the survey or questionnaire used in the study and ensure that you have provided sufficient details that others could replicate the analyses.

For instance, if you developed a questionnaire as part of this study and it is not under a copyright more restrictive than CC-BY, please include a copy, in both the original language and English, as Supporting Information.

4. You have mentioned that the questionnaire was pre-tested, but not whether it was validated.

Please clarify if it was validated. If this did not occur, please provide the rationale for not validating the questionnaire.

Reviewers' comments:

Reviewer's Responses to Questions

**Comments to the Author**

1. Is the manuscript technically sound, and do the data support the conclusions?

Reviewer #1: Yes

Reviewer #2: Yes

Reviewer #3: Yes

2. Has the statistical analysis been performed appropriately and rigorously? 

Reviewer #1: Yes

Reviewer #2: Yes

Reviewer #3: Yes

3. Have the authors made all data underlying the findings in their manuscript fully available?

Reviewer #1: Yes

Reviewer #2: Yes

Reviewer #3: Yes

4. Is the manuscript presented in an intelligible fashion and written in standard English?

Reviewer #1: Yes

Reviewer #2: Yes

Reviewer #3: Yes

5. Review Comments to the Author

Reviewer #1: The paper is about “financial hardship of diabetic care” which is an important topic as it is known and shown in the literature that financial hardship(risk) protection is the international and national agendas, such as SDG 3.8.2 and the Ethiopia’s health sector’s fifteen years strategic plan (2015-2035). One of the barrier to achieve universal health coverage by 2030 is financial barrier to get quality of health care, mainly in low income countries like Ethiopia. So it is important to know about the Ethiopian diabetic patients’ financial hardship protection status and its associated factors in Ethiopia to inform policy options in the country in order to attain the universal health coverage without financial hard.

Despite about it, there are some comments that need to be addressed before this paper could be published.

Comments:

1. Add the response rate in the abstract section of manuscript in the result part

2. In the introduction section of the manuscript, there are long sentences that can be broken down to simple, short and concise sentences as per the plos one journal recommendation. For instance, “It is a key health system objective and one of the target in the Sustainable Development Goals (SDGs), specifically SDG3, and Health Sector Transformational Plan (HSTP) of Ethiopia” can be broken down to two short and concise sentences.

3. On page 19, there is empty page, it should be omitted

4. The title of table 6 should be edited and why the authors make the footnote of table 6 bold?

5. The authors should add the questionnaire as supporting information for the manuscript

Reviewer #2: General comment

• The manuscript has presented an original work and has applied the techniques and requirements of a scientific study with strong conceptual and statistical analysis. It is publishable with some finer correctable suggestions forwarded here under step by step.

Simple Notice

• Please try to cancel the repeatedly used pronoun ‘we’ as a doer. The interest should be expressing on what has been done.

• In the conclusion part it is good if you state in such a way that Economic burden of DM care is stronger among the less privileged populations: the less educated, the poorest and unemployed. Because your regression result shows that.

Introduction:

• Reconstruct some lengthy sentences to make it simple and clear. Eg. In the second paragraph the last sentence and the second sentence in the last paragraph

Methods and Materials

Study Design and participants

• Cancel the phrase ‘Institutional based” because later on you mentioned that the study is conducted on hospitals. By implication the readers can understand that the study was institutional based. Don’t write the knowns.

• Mention whether you included both outpatients and inpatients or either of the two. Since DM care is part of CBHI benefit package, please declare that whether you included CBHI enrolled patients or not? If you included them, in the cost calculation you need to specify if you calculate the cost regardless of who pays or if you included only the OOP? You can use it as eligibility criteria.

Sample size and Sampling Method

• Make the sampling clear by mentioning how many samples you took from how many populations in both of the hospitals. It is good to specify sampling technique rather than broadly saying random technique: what specific random technique was used to recruit study participants. In fact, in your abstract you said it (simple random method), but still if possible, it is advisable to be very specific (What specific simple random method?).

Measuring catastrophic health expenditure

• Rather than saying health expenditure (which is more general), you need to say expenditure to diabetic care (your case). A curiosity question is that why did you represent expenditure to health care with Ti? What if you change it to Hi (H to mean health or DMxi- expenditure of the ith patient to receive care for DM) Then the formula would be

o [DMxi/(xi-Fxi)] *100 (look the formula again. It seems that 100 multiplies only the denominator).

• If you already used a threshold of 40%nonfood expenditure, what is the importance of the last sentence?

Measuring incidence and intensity of catastrophic payments

• Hopefully by saying ‘z’ you are referring to the threshold (40% of nonfood expenditure)!!! If that is the case say so. Otherwise mention what it is.

• Generally, the three formulas need to be more detail. Simply formula one is about the ratio of patients who experienced CHE? Is it that much important to sophisticate the formula?

• Is Ei in formula 2 similar with the one inf formula 1? – Don’t you think that this will positively inflate the overshoot? What about the negative overshoot?

Results

Socio-demographic and Socioeconomic……

• If you can it would be more informative to report mean income of patients. Because it would help the reader to compare it against the mean monthly total expenditure.

• In table 1- better to cancel the variable ’Sex’. Since the males are reported in the text the reader understands that the other share is for females. The same is true for Religion and Residence. Narrate the dichotomy variables in the text and avoid including it in the table. This helps the table to be smart.

Cost of diabetes mellitus treatment, household expenditure

• Better to mention how the mean monthly expenditures and costs were calculated. (You need to see the numerator and denominator).

Coping Strategies

• Correct the last two sentences. Grammatically they are not correct.

• Put it after the section next to it (Catastrophic health expenditure and impoverishment).

Catastrophic health expenditure and impoverishment

• Reporting these variables with its classification based on different variables do not add value. That means table 3 and majority of the preceding texts can be canceled. Because you are going to fit regression model for it, which can be more informative that this table.

Discussion

• In the 5th paragraph rather than using comma use brackets so separate the numbers you summed up. That is instead of putting as……. patients included in the study had invested 63.46%, 23.46%+40%, of the…….. write it as ……patients included in the study had invested 63.46% (23.46%+40%) of the…. . Moreover, you should see the interpretation of catastrophic overshoot and MPO again. The specified expenditures are for percentage of nonfood expenditure rather than percentage of total expenditure.

• Paragraph six- Is it fair to make that 5% impoverishment is similar with 24%, 19.1%, 29.9%? see it again.

• The four factors affecting CHE in your study indicated that ’those who were worse off socio economically are more prone to catastrophic expenditure than that of the counterparts.” This should be discussed very well.

Conclusion

• The last sentence should be specific. Specifically, the recommended protection mechanisms should target the have-nots like the poor, the unemployed and uneducated since the financial hardship is stronger for them than the other half.

Reference

• See some referencing styles. For example, reference number 27 can be rewritten in a better way using standard referencing style.

Reviewer #3: Financial hardship of diabetic careis the topic of this article which is a burning issue in both the global and in the Ethiopian contexts as the authors described in the introduction section of the manuscript.But as the best search of the reviewer, there is limited evidence about it in Ethiopia. The results of this study can be used as an evidence to design evidence based intervention in Ethiopia to protect diabetic patients from financial hardship, to ensure equity health care in the country. Therefore, this article is important and relevant to policy makers of Ethiopia’s health sector to accomplish Universal Health Coverage (UHC) by the year 2030.

However, the reviewer recommends the following minor revisions in the manuscript before publication.

In the introduction section: In the manuscript, some of the sentences are long which should be short as much as possible so that the potential readers will understand the paper easily.

o In the methods section of the manuscript, Describe your study area how many public hospitals found in Bahir Dar city administration?

o What is your sampling method? describe it clearly

Result section

o use appropriate punctuations in some of sentences and correct the editorials for language

o on table 1 'Amara' or 'Amhara'?

o make Table 5 clear in meaning and drawing as well

6. PLOS authors have the option to publish the peer review history of their article (what does this mean?). If published, this will include your full peer review and any attached files.

Reviewer #1: No

Reviewer #2: No

Reviewer #3: No

---

## [Author Response · Author response to Decision Letter 0]

11 Dec 2020

Dear reviewers and editor,

Thank you so much for your constructive comments to improve our manuscript. 

Below are our point by point responses for the comments

Responses for the academic editor

1. We have adhered to the requirements of PLOS ONE's style as much as possible.

2. We, the authors, and our colleague (Dr.Walelign Kindie, Email: walelignkindie2@gmail.com, an academic staff of Bahir Dar University, Tel: +2519 38883386) thoroughly copyedited the manuscript for language usage, spelling, and grammar.

3. We, the authors, submitted the local language, Amharic, and English versions of questionnaire tools as additional files.

4. We have conducted pretest before survey as stated on page 6 line number 106 in the manuscript. The purpose of the pretest was to ensure that the understandability and clearness of the questionnaire for the respondents and it was modified accordingly. We modified the flow, order, skip patterns and the allocated time for interview in the questionnaire. The questionnaire was developed based on previous validated questionnaires (standard one) that is why validation is not done for our study.

Responses for the comments (reviewer 1)

1. We added the response rate in the abstract section of manuscript in the result part on page 2 line number 37.

2. In the introduction section of the manuscript, long sentences have been broken down to simple, short and concise sentences as per the comments. For instance the long sentence such as, “It is a key health system objective and one of the target in the Sustainable Development Goals (SDGs), specifically SDG3, and Health Sector Transformational Plan (HSTP) of Ethiopia” were broken down to two short and concise sentences: It is a key health system objective and one of the targets in the Sustainable Development Goals (SDGs), specifically SDG3. Financial risk protection is also the target of Health Sector Transformational Plan (HSTP) of Ethiopia (on page 4 line numbers 54-57).

3. In the previous manuscript ,there is empty page, it was omitted right now

4. The title of table 6 was edited and its footnote was unbold?

5. The authors added the questionnaire as supporting information in the current manuscript.

Responses for the comments (reviewer 2)

1. We used pronoun ‘we’ as a doer as the reviewer has commented on the manuscript. Even though, the interest is expressing on what has been done, to make it clear the doer is also important as per the norm of scientific writing. 

2. In the conclusion part of the abstract we stated in such a way that economic burden of DM care is stronger among the less privileged populations: the less educated, the poorest and unemployed as per the recommendation of the reviewer on page 3 line numbers 43-44.

3. We edited lengthy sentences to make it simple and clear throughout the manuscript. 

4. In the methods part the phrase “institutional based “was used to indicate the type of study design that we used to answer our research question. The readers (those who are not expertise in epidemiology) might not understand the type of study design by reading the setting of the study as reviewer stated.

5. We, the authors, included patients who are not CBHI enrolled as stated on page 6 line numbers 91-93 in the manuscript.

6. We, the authors, mentioned (on page 6 line numbers 100-102) how many samples we took from how many populations in both of the hospitals. We recruited 325 diabetic patients from Felege Hiwot comprehensive specialized hospital and 97 diabetic patients from Addis Alem primary hospital by using simple random sampling method with computer generated random number.

7. We, the authors, replaced health expenditure to expenditure of diabetic care in the measuring catastrophic health expenditure part of the manuscript. Moreover, we used DMxi instead of Ti. - Expenditure of the ith patient to receive care for DM. All formulas were edited as per the comments of reviewer.

8. We used a threshold of 40% nonfood expenditure for analytical analyze, however, we also reported the results at different recommended levels: varied between 10 and 40% of threshold level.

9. We elaborated more about the overshoot in the current version.

10. In the result part, we reported the monthly income in the table in the previous version. As per the comment of the reviewer, we have stated the mean monthly income in the current version of the manuscript on page 12 line numbers 207-208. 

11. In table 1- the reviewer recommended that “better to cancel the variable ’Sex’. Since the males are reported in the text the reader understands that the other share is for females. The same is true for Religion and Residence. Narrate the dichotomy variables in the text and avoid including it in the table. This helps the table to be smart”. However, if we cancel these important variables, the table may not be self-explanatory or self-contained that leads to ambiguity due to the incomplete information in the table. 

12. We have mentioned how the mean monthly expenditures and costs were calculated in the method of costing part starting from on page 7 line number 119.

13. Regarding to table 3 in the result part, the table informs the readers the proportion of catastrophic diabetic care expenditure and impoverishment due to diabetic care in each category, more specifically those variable not reported in the regression table. Hence, it adds value for the readers.

14. We have edited and corrected the discussion as per the recommendation of the reviewer’s comments on page 23 line numbers: 334,336 and 340.

15. In conclusion part, we have specified the last sentence as per the reviewer’s comment on page 25 line numbers 372 and 373.

16. We have edited the references list in the manuscript.

Responses for the comments (reviewer 3)

1. In the introduction section, we edited the long sentences in the current version of abstract.

2. In the method section, we described about the study area in which how many public hospitals found in Bahir Dar city administration on pages 5&6 line numbers 85-90.

3. Regarding to sampling method, we stated about it on page 6 line number 99-102.

4. In the result part, we and our colleague have copy edited the whole manuscript including result part.

---

## [Decision Letter · Decision Letter 1]

11 Jan 2021

Are Ethiopian diabetic patients protected from financial hardship?

PONE-D-20-25611R1

Dear Dr. Tsega,

We’re pleased to inform you that your manuscript has been judged scientifically suitable for publication and will be formally accepted for publication once it meets all outstanding technical requirements.

Kind regards,

Khurshid Alam, Ph. D.

Academic Editor

PLOS ONE

Additional Editor Comments (optional):

Reviewers' comments:

Reviewer's Responses to Questions

**Comments to the Author**

1. If the authors have adequately addressed your comments raised in a previous round of review and you feel that this manuscript is now acceptable for publication, you may indicate that here to bypass the “Comments to the Author” section, enter your conflict of interest statement in the “Confidential to Editor” section, and submit your "Accept" recommendation.

Reviewer #3: All comments have been addressed

2. Is the manuscript technically sound, and do the data support the conclusions?

Reviewer #3: Yes

3. Has the statistical analysis been performed appropriately and rigorously? 

Reviewer #3: Yes

4. Have the authors made all data underlying the findings in their manuscript fully available?

Reviewer #3: Yes

5. Is the manuscript presented in an intelligible fashion and written in standard English?

Reviewer #3: Yes

6. Review Comments to the Author

Reviewer #3: The results of this study can be used as an evidence to design evidence based intervention in Ethiopia to protect diabetic patients from financial hardship, to ensure equity health care in the country. Therefore, this article is important and relevant to policy makers of Ethiopia’s health sector .

I found that all comments were provided are corrected.

7. PLOS authors have the option to publish the peer review history of their article (what does this mean?). If published, this will include your full peer review and any attached files.

Reviewer #3: No

---

## [Editor Report · Acceptance letter]

13 Jan 2021

PONE-D-20-25611R1 

Are Ethiopian diabetic patients protected from financial hardship? 

Dear Dr. Tsega:

I'm pleased to inform you that your manuscript has been deemed suitable for publication in PLOS ONE. Congratulations! Your manuscript is now with our production department. 

Kind regards, 

on behalf of

Dr. Khurshid Alam 

Academic Editor

PLOS ONE